**Cite this article:** luo Y, Yang L, Liu Q, Yan Y. 2021 *In situ* polyaniline coating of Prussian blue as cathode material for sodium-ion battery. *R. Soc. Open Sci.* **8**: 211092. https://doi.org/10.1098/rsos.211092

materials science/nanotechnology

*in situ* coating, Prussian blue, ball-milling, battery

**Author for correspondence:**
Youwei Yan
e-mail: yanyw@hust.edu.cn

This article has been edited by the Royal Society of Chemistry, including the commissioning, peer review process and editorial aspects up to the point of acceptance.

# *In situ* polyaniline coating of Prussian blue as cathode material for sodium-ion battery

## Yu luo, Lingxiao Yang, Qing Liu and Youwei Yan

State Key Laboratory of Material Processing and Die and Mould Technology, School of Materials Science and Engineering, Huazhong University of Science and Technology, Wuhan, Hubei 430074, People's Republic of China

YI, 0000-0002-1727-2307

Prussian blue (PB) has great potential for use as a sodium cathode material owing to its high working potential and cube frame structure. Herein, this work reports a two-step method to synthesize PB with ascorbic acid as the ball-milling additive, which improves the electrochemical rate performance of PB during the traditional co-precipitation method. The obtained PB sample exhibited a superior specific capability (113.3 mAh g$^{-1}$ even at 20 C, 1 C = 170 mA g$^{-1}$) and a specific capacity retention of 84.8% after 100 cycles at 1 C rate. In order to enhance the cycling performance of the PB, an *in situ* polyaniline coating strategy was employed in which aniline was added into the electrolyte and polymerized under electrochemical conditions. The coated anode exhibited a high specific capacity retention of 62.7% after 500 cycles, which is significantly higher than that of the non-coated sample, which only remains 40.1% after 500 cycles. This development has shown a great potential as a low-cost, high-performance and environment-friendly technology for large-scale industrial application of PB.

## 1. Introduction

In recent years, sodium-ion batteries (SIBs) have gained attention as potential substitutes for Li-ion batteries due to the abundance and low cost of sodium [1,2]. However, the slow kinetics and poor cycling stability resulting from the size of Na$^+$ (larger than Li$^+$) significantly limit the application of SIBs [3,4]. Prussian blue (PB), with an open framework and stable structure, can meet the requirements of high power density and long cycle life as a cathode material of SIB [5]. However, PB synthesized via a traditional co-precipitation method is time-consuming and the resultants show a low electric conductivity and a low specific

capacity at a high rate (20 C less than 100 mAh g$^{-1}$, 1 C = 170 mA g$^{-1}$) [6,7]. These disadvantages limit the commercial application of PB. Thus, it is necessary to develop an alternative method and optimize the synthetic process. Carbon coating via the pyrolysation of organic precursors at a high temperature is considered as an effective way to enhance the conductivity of electrode materials. Unfortunately, this method is not applicable to PB due to its instability over 250°C [8]. Conductive polymers are used to modify the surface of PB via the solution method, but the growth way is uncontrollable, thus it is difficult to generate an intact conducting polymer coating layer on the surface of PB [9,10]. Instead of these defective and complex approaches, PB can be synthesized via a ball-milling method which can improve its high rate performance [11]. In this work, the precursor of PB was treated via a ball-milling process followed by rapid precipitation, and the usability of ascorbic acid (AA) in the ball-milling process was studied. In order to enhance the cycling performance of the PB, an *in situ* polyaniline (PANI) coating strategy was employed in which aniline was added into the electrolyte and polymerized under electrochemical conditions. It is proved that the PB synthesized in this method has lower interstitial water content compared with the traditional synthesis method, and the electrochemical rate performance of the PB greatly improved owing to the large surface area. Besides, AA can prevent PB oxidized (Fe$^{2+}$ changed into Fe$^{3+}$) during the ball-milling process. In addition, aniline used as an electrolyte additive for polymerization makes the sample more even and thicker compared with the previously reported work, and the conductive polymer coat further improves the electrochemical cycle performance and electronic conductivity. This development has shown a great potential as a low-cost, high-performance and environment-friendly technology for large-scale industrial application of PB.

## 2. Results and discussion

The method of synthesizing Sample-A, Sample-O and Sample-L is given in the synthesis method section. Sample-A refers to the PB produced by two-step synthesis method used AA as additive. Sample-O is defined for the PB produced by two-step synthesis method without any additives. And Sample-L refers to the PB produced by mixing raw material with directly rapid co-precipitation, without ball-milling. Figure 1*a* shows that the main PB peaks of Sample-A, Sample-O and Sample-L (JCPDS no. 1-0239). Sample-A shows the stronger intensity of characteristic peaks than others, which is previously reported that the intensity of the X-ray diffraction (XRD) peaks increases with increasing precursor concentration, temperature and thickness [12–15]. Electronic supplementary material, figure S1a shows the XRD patterns of Sample-A-precursor and Sample-O-precursor. The main peaks of the two samples can be attributed to Na$_4$Fe(CN)$_6$ and some impurities. The peak of the PB phase is scarcely observable [12,16,17]. The difference between figure 1*a* and electronic supplementary material, figure S1a in XRD patterns indicates that most of the PB phase in Sample-A and Sample-O are formed in the process of rapid precipitation. The peaks related to the AA of Sample-A-precursor are revealed by the Fourier transform infrared spectroscopy (FT-IR) result as shown in the electronic supplementary material, figure S1b [6]. The figure 1*b* shows the FT-IR result, revealing that all the samples have a strong peak at 2083 cm$^{-1}$, which corresponds to the C-N groups. The plot of all the samples shows a peak in the 1607 cm$^{-1}$, which corresponds to the interstitial water, and the peak of Sample-L in this position is stronger than others, which indicates that the mixed raw material without ball-milling leads to more interstitial water content. The thermo-gravimetric curves in figure 1*d* further prove that. The wide spectra of X-ray photoelectron spectroscopy plots for all the samples are shown in figure 1*c*, and the narrow spectra of Fe 2p for all the samples are shown in the electronic supplementary material, figures S1c, S1d and S1e. The relative peak area ratio of N-Fe$^{III}$/N-Fe$^{II}$ of Sample-O is higher than that of the other two samples. The XPS result shows that the AA can prevent Fe$^{2+}$ from being oxidized to Fe$^{3+}$ to some extent during the ball-milling process. The effect of AA as an additive in the conventional co-precipitation method was figured out in other literature [6]. Through the calculation according to the thermo-gravimetric analyses (TGA) curve (figure 1*d*) and ICP/elemental analysis (EA) result (electronic supplementary material, table S1), the formula of Sample-A is determined to be Na$_{1.515}$Fe[Fe(CN)$_6$]$_{0.879}$·2.7H$_2$O. The calculation process of the exact formula is listed in the electronic supplementary material information.

The morphology of all the samples observed by scanning electron microscopy (SEM) images and the transmission electron microscopy (TEM) images are shown in electronic supplementary material, figures S2 and S3. The results indicate that Sample-A and Sample-O consist of irregular particles rather than the cubes in Sample-L, and the particles are in nanoscale size range. It is reported that the nanoscale size of

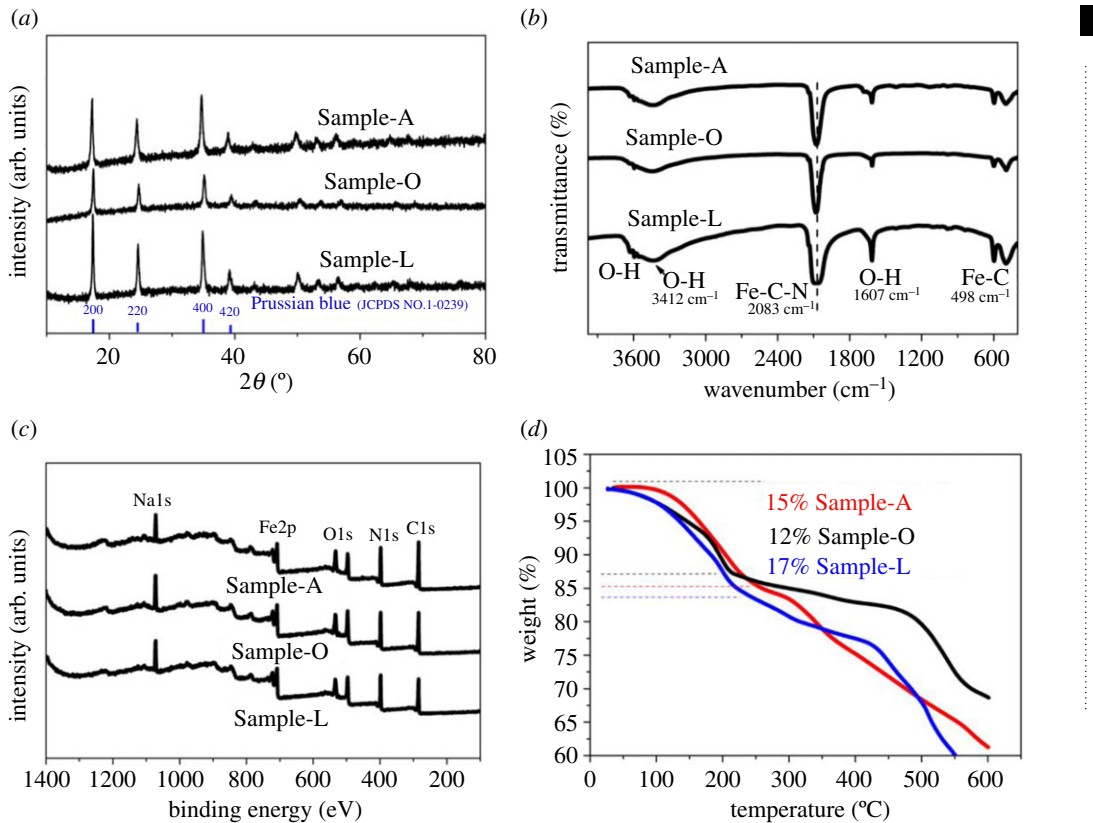

**Figure 1.** (a) XRD patterns of all the samples. (b) FT-IR spectra of all the samples. (c) XPS wide scan survey spectrum of all the samples. (d) TGA curves of all the samples.

the particles can improve battery performance [18]. The Brunauer–Emmett–Teller (BET) plot in electronic supplementary material, figure S4 and the Barrett–Joyner–Halenda (BJH) plot in electronic supplementary material, figure S5 reveal that the specific surface area of Sample-A is $94.26 \, \mathrm{m^2 \, g^{-1}}$ and the pore size of this sample is mainly distributed at 2.87 nm. The specific surface area of Sample-O is $56.16 \, \mathrm{m^2 \, g^{-1}}$ and the pore size of this sample is mainly distributed at 3.35 nm. The specific surface area of Sample-L is $39.44 \, \mathrm{m^2 \, g^{-1}}$ and the pore size of this sample is mainly distributed at 2.73 nm. Compared with the cubic ones, the irregular particles deliver a larger surface area. The larger surface area would in turn increase the contact area between the electrolyte and the cathode material and therefore improve battery performance. Such finding can be found in other previous reports [19].

Figure 2a shows the initial charge/discharge profiles of Sample-A, Sample-O and Sample-L at a current density of 1 C. The increase in specific capacity in the first few cycles is caused by the significant polarization of the original material, which prevents the full utilization of the electrode [20]. Towards the anode materials, this phenomenon could be owing to the formation of a solid electrolyte interface, as found by other researchers [21]. In subsequent cycles, the specific capacity increases with the gradual activation of the electrode material, whereby the contact between active materials particles and the conductive agents is improved. Sample-A, Sample-O and Sample-L electrodes show the initial discharge specific capacities of 129.6, $113.1 \, \mathrm{mAh \, g^{-1}}$ and $118.7 \, \mathrm{mAh \, g^{-1}}$, and coulombic efficiency of 87.1%, 86.6% and 86.17%, respectively. This is further displayed in figure 2a. Two typical voltage plateaus can be observed in the discharge curves, referring to the $Fe^{III}/Fe^{II}$ redox couple, and the third couple contributes little platform capacity [7]. Rate performance tests were performed under various current densities from 1 C to 20 C, as displayed in figure 2b. At 1, 2, 5 and 20 C, Sample-A delivered specific capacities of 128.1, 120.7, 120.2 and $113.3 \, \mathrm{mAh \, g^{-1}}$, respectively. However, under the same current densities, Sample-O achieved the specific capacities of 117.4, 112.3, 111.7 and $102.9 \, \mathrm{mAh \, g^{-1}}$. At 1, 2, 5 and 20 C, Sample-L achieved the specific capacities of 115.7, 109.4, 105.6 and $94.3 \, \mathrm{mAh \, g^{-1}}$. The difference between Sample-O and Sample-L in a high rate performance (20 C, 1 C = $170 \, \mathrm{mA \, g^{-1}}$) indicates the advantage of a large surface, corresponding to the BET result. The difference between Sample-A and Sample-O in rate performance indicates the advantage of AA, owing to the ion pathway created inside of the particle. This improvement is also reported in other studies [6]. Compared with the traditional synthesis method, our method noticeably improves the rate performance of the electrode

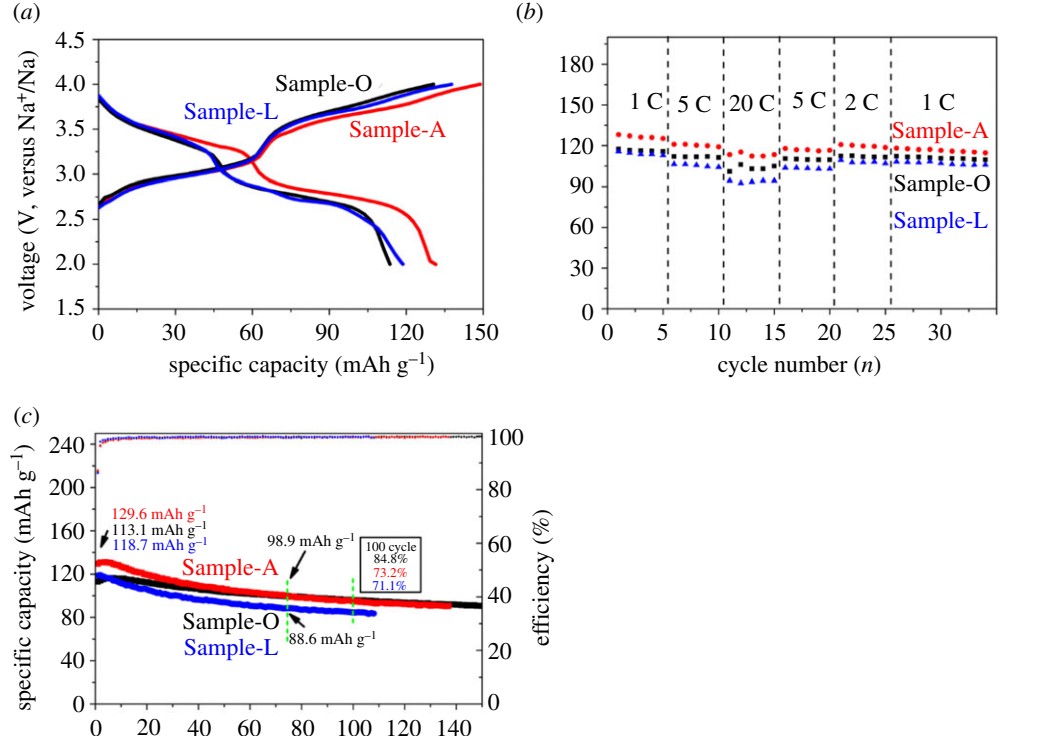

**Figure 2.** Electrochemical performances of all the samples. (*a*) The galvanostatic initial charge/discharge plots between 2.0 and 4.0 V at a current density of 1 C (1 C = 170 mA g$^{-1}$); (*b*) electrochemical rate performance; (*c*) electrochemical cycle performance at 1 C from 1 to 150 cycles (1 C = 170 mA g$^{-1}$).

material [22]. Figure 2*c* shows the electrochemical cycle performance at a current density of 1 C (1 C = 170 mA g$^{-1}$); Sample-A, Sample-O and Sample-L electrodes achieve 129.6, 113.1 and 118.7 mAh g$^{-1}$, thus maintaining 73.2%, 84.8% and 71.1% of the initial specific capacity values after 100 cycles, respectively. Figure 2*c* shows that after 76 cycles, Sample-O achieved a specific capacity of 98.9 mAh g$^{-1}$, indicating that the cycle stability of Sample-A can be improved. This phenomenon has been explained in other studies [6]. The electronic supplementary material, figure S12 summarized the difference of PB produced by this work and others work reported [7,23–31].

To further enhance the ionic conductivity and improve the electrochemical cycle stability of Sample-A, aniline was employed as an electrolyte additive. In total, 0.3 wt% of aniline was added into the electrolyte. The electrolyte contained NaClO$_4$ dissolved in 1:1 (v/v) diethyl carbonate/ethylene carbonate, with 5 wt% fluoroethylene carbonate. To oxidize the polymerization of aniline, an activation process was established for the coating of aniline on the electrode after the battery was assembled. Figure 3*a* shows the CV curves of Sample-A and Sample-A-0.3% after the activation process and a couple of small peaks at 3.45/3.23 V. This phenomenon might be caused by the extraction/insertion of Na$^+$ from some other sites in the structure, and corresponds to the XPS result related to the N-Fe$^{III}$ [7,23]. The same peak position of Sample-A-0.3% reveals smaller than that at Sample-A, which might be due to the oxidation environment caused by the polymer. Figure 3*b* displays the cycling performance of Sample-A and Sample-A-0.3% after the same activation process. At a current density of 1 C, Sample-A and Sample-A-0.3% electrodes maintained 40.1% and 62.7% of their initial specific capacity after 500 cycles, respectively. Figure 3*c* shows the rate performance of Sample-A-0.3% at various current densities from 0.2 to 20 C. At 0.2, 1, 2, 5 and 20 C, Sample-A-0.3% achieved the specific capacities of 156.5, 149.9, 145.9, 140 and 125.6 mAh g$^{-1}$, respectively. At 20 C, the electrode delivered a reversible specific capacity of 125.6 mAh g$^{-1}$, thus maintaining 80.3% of the specific capacity at 0.2 C. Figure 3*d* further illustrates the charge/discharge curves at different current densities. As a conductive polymer, PANI can be uniformly coated onto the material through the activation process. After the activation process, the electrochemical efficiency of the electrode reaches 98%. This method improves PB's poor electron and ion conductive capability [32].

The electrochemical impedance spectroscopy (EIS) results support the conclusion and show that the charge transfer resistance of Sample-A-0.3% is noticeably lower during the activation process (electronic

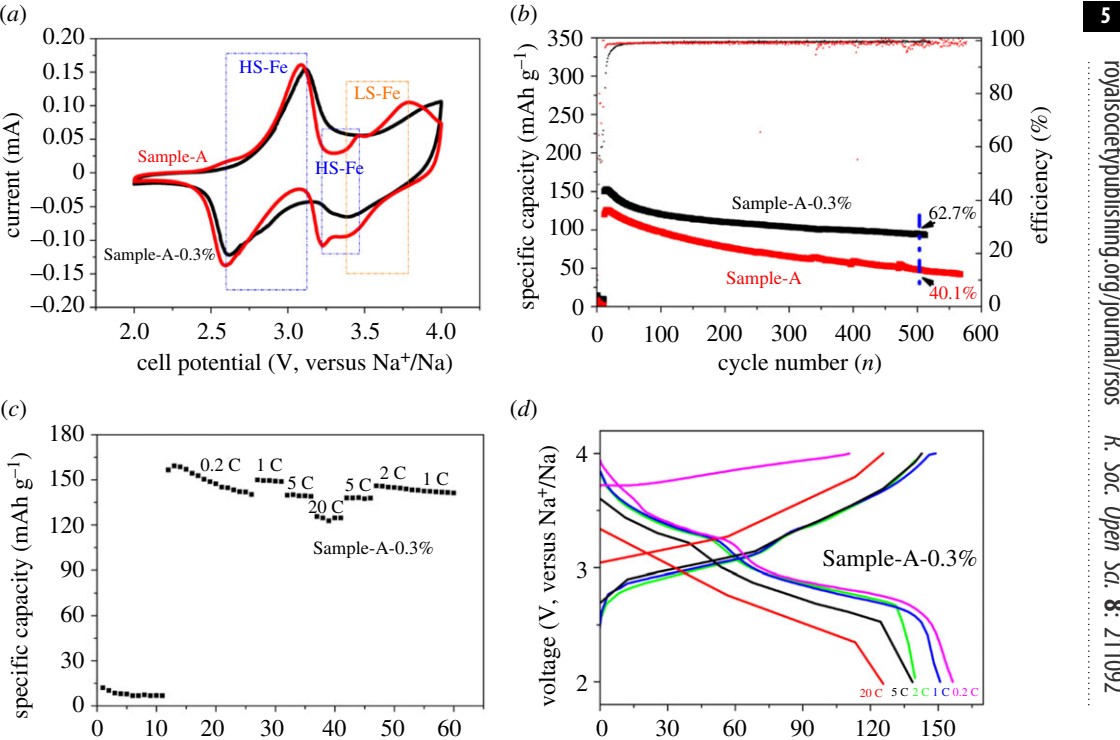

**Figure 3.** (a) Cyclic voltammetry curves of Sample-A and Sample-A-0.3% after the activation process. (b) Electrochemical cycle performance of Sample-A and Sample-A-0.3% at 1 C (1 C = 170 mA g$^{-1}$). (c) Electrochemical rate performance of Sample-A-0.3%. (d) Charge/discharge curves of Sample-A-0.3% at different rates.

supplementary material, figure S6). The gradually changed radius of the semicircle (related to the Rct) represents the gradually enhanced conductivity. The charge transfer kinetics of the electrodes benefitted from the conductive polymer coating and a more stable interface between the electrode and electrolyte in the running cycles. Similar phenomena have also been observed in the other literature [33,34]. This result shows that PANI can reduce the electrical resistance of the electrochemical process. The increase in discharge specific capacity is rarely caused by aniline. Pure carbon was used as the anode electrode in the same electrolyte with Sample-A-0.3%, which is named as Sample-P (electronic supplementary material, figure S7a), and the aniline provided little discharge specific capacity. The cathode active material was coated on the Al foil, as aluminium did not contribute to the measured capacity [35]. Electronic supplementary material, figure S7b shows the results of Sample-A-3%, using the 3 wt% aniline as an electrolyte additive, and the electrochemical performance results were incorrect. Electronic supplementary material, figures S7c and S7d display the electrochemical performance of Sample-A and Sample-A-0.3% with a low carbon content in the electrode (only 2.5 wt % carbon content in the electrode, PB: Super P: Polyvinylidene fluoride (PVDF) = 87.5 : 2.5 : 10). Even when the carbon content dropped from 20 to 2.5 wt%, Sample-A-0.3% still delivered a discharge specific capacity of 136.8 mAh g$^{-1}$ at 1 C and retained a 40% specific capacity after 500 cycles owing to the homogeneous PANI coating. Electronic supplementary material, figure S8a shows the XRD pattern of the pristine Sample-A-0.3%'s electrode plate, Sample-A-0.3%'s electrode plate after the activation process, and Sample-A-0.3%'s electrode plate after the activation process and the 500th cycle electrochemical test. The results indicate that the structure of the sample does not change after aniline coating. Additionally, this coating improves the structure's stability. The Raman test results displayed in electronic supplementary material, figure S8c also prove this. As the electrolyte was added at a low dose, the carbon in the electrode shows a strong peak, but the aniline peaks are not obvious in the Raman test. Electronic supplementary material, figure S8b shows the XRD pattern of the PANI which indicates that PANI is amorphous [36,37]. The ex situ FTIR result shows that the aniline polymerized on the PB during the activation process (electronic supplementary material, figure S8d) [9]. The ex situ SEM images of Sample-A and Sample-A-0.3% electrode plate section shown in the electronic supplementary material, figure S9 reveal that the PANI coating can reduce variation of the electrode plate, which improve the electrochemical cycle performance.

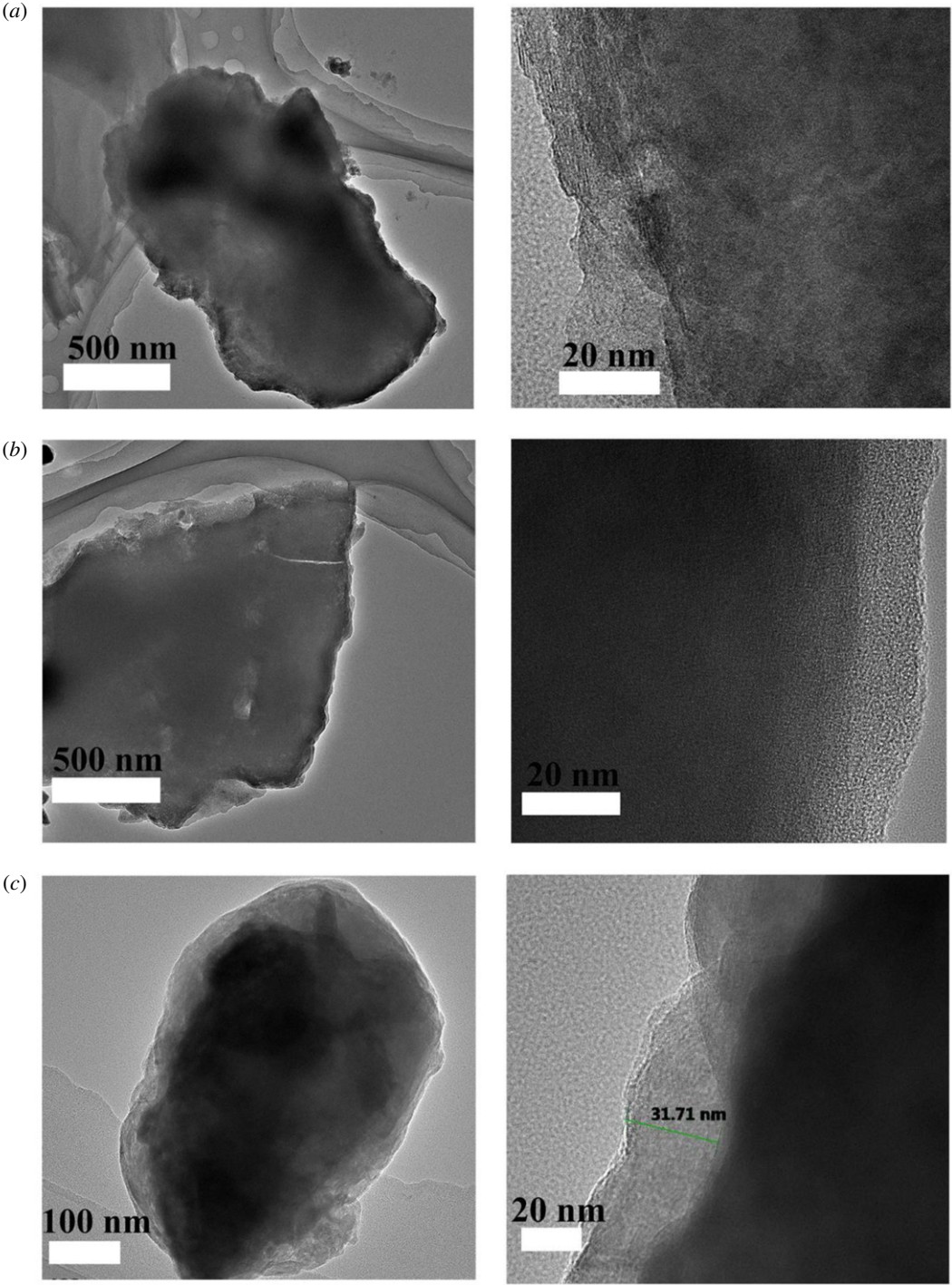

**Figure 4.** Ex situ TEM of Sample-A-0.3% in the activation process: (*a*) at the 3rd cycle; (*b*) at the 7th cycle and (*c*) at the 12th cycle.

To further explore the coating and polymerization phenomena occurring during the activation process, ex situ TEM characterization was conducted. Figure 4 shows the PB particle being gradually coated with aniline; the coating was approximately 30 nm thick after the activation process. Such nanoscale size of the particles is important in improving battery performance [18]. The conductive polymer coating was thicker and more even during the activation process than in the synthesis processes [9]. Electronic supplementary material, figure S10 shows the ex situ XRD test result for Sample-A-0.3%. This test reveals the structural evolution during the discharge/charge process of Sample-A-0.3% after the activation process. The electrode was charged from 3.0 to 4.0 V and then discharged from 4.0 to 2.0 V. As the charge voltage rose, the positions of the 220 and 400 characteristic peaks in the XRD pattern moved to higher positions. This phenomenon is associated

with $Na^+$ ion extraction [38,39]. As the discharge voltage declined, the 220 and 400 characteristic peaks in the XRD pattern moved to lower positions. This phenomenon corresponds with the $Na^+$ ion insertion. The ex situ XRD pattern shows that the crystalline structure of Sample-A-0.3% is pretty stable after the activation process. As shown in electronic supplementary material, figure S11, the ex situ XPS results further prove that when the electrode was charged to 4 V, the peaks related to $Fe^{3+}$ revealed, which was corresponding to $Na^+$ extraction. As the electrode was discharged to 2 V, the peaks related to $Fe^{2+}$ revealed, corresponding to $Na^+$ insertion. This proves the material was stable [40]. The enhancement of the rate and cycling performance of Sample-A-0.3% are due to the effect of PANI coating, thus improving the ion and electronic conductivity. Furthermore, the conductive polymer coating layer can suppress the side reactions at high voltages, thus further enhancing the structure's stability in the electrochemical process. The electrochemical efficiency of the polymer coating in the first few cycles is pretty low, which needs to be further investigated in the future [41].

# 3. Conclusion

As compared with the traditional synthesis method, the PB synthesized via a ball-milling process followed by rapid precipitation delivers a great electrochemical performance, owing to the lower interstitial water content and the larger surface area. AA was used as a ball-milling additive in the synthesis of PB, and the specific capacity of the obtained PB used as the cathode material in the half cell reached 129.6 mAh g$^{-1}$ at 1 C, with an initial coulombic efficiency of 87.1%. Sample-A showed a good electrochemical specific capacity and electrochemical rate performance compared with Sample-O, because the AA massively improves the degree of crystallinity of PB. In addition, aniline was used as an electrolyte additive to further enhance the electrochemical performance of PB. Sample-A-0.3% achieved a much better electrochemical cycle performance and rate performance than Sample-A after the activation process. The even coating of PANI hugely improves the material's ion conduction ability and electronic conductivity. Such a method to improve the electrochemical properties of PB is simple and effective, which will guide further research on the application of PB as a cathode material.

# 4. Experimental section

## 4.1. Materials

Ferrous sulfate (99% AR, Sigma-Aldrich), sodium ferro-cyanide (99% AR, Sigma-Aldrich), aniline, AA (99.5% RT, Sigma-Aldrich) and anhydrous ethanol solution (ACS, ≥99.5%, Sigma-Aldrich) were used. Deionized water was used in all the synthesis process.

## 4.2. Synthesis of the cathode samples

### 4.2.1. Sample-A

Four millimoles ferrous sulfate and 6 mmol sodium ferro-cyanide were heated to remove the adsorbed water before using and were then subjected to a ball-milling treatment (ball to powder weight ratio was 2 : 1, rotation speed was 250 r.p.m. for 2 h) with a specific amount of AA (3 mmol), and the ball-milled mixture denotes as the precursor (the above-mentioned part was the ball-milling process). Next, the precursor was transferred to a beaker with the appropriate amount of deionized water (100 ml). The obtained suspension was stirred for a period of time (5 min). After that, the suspension was centrifuge washed (the centrifugal speed was 5000 r.p.m. for 3 min) with deionized water (100 ml) three times and anhydrous ethanol (100 ml) three times in sequence, followed by drying in a vacuum oven at 80°C (higher drying temperature can make the material have less interstitial water content, but in this stage, the loss of interstitial water affects structural stability) to obtain the final sample (this mentioned part was the rapid precipitation process). The final sample was labelled Sample-A (A was the capital letter of the word 'ascorbic acid').

Sample-O (O was the capital letter of the word 'origin') was synthesized in a similar way like Sample-A, excepting that no AA was added in the ball-milling process.

Sample-L (L was the capital letter of the word 'liquid'), the mixed sample (4 mmol ferrous sulfate, 6 mmol sodium ferro-cyanide and 3 mmol AA), directly synthesized the PB by rapid precipitation process (without ball-milling process), as for the blank comparison.

## 4.3. Electrolyte additive samples

### 4.3.1. Sample-A-0.3%

The cathode electrode (details in the Electrochemical measurements section) was the same as that used for Sample-A, but 0.3 wt% aniline was added into the electrolyte for conductive polymer coating. The electrolyte was $NaClO_4$ dissolved in 1:1 diethyl carbonate/ethylene carbonate containing 5 wt% fluoroethylene carbonate.

### 4.3.2. Sample-A-3%

Sample-A-3% was similar to Sample-A excepting the 3 wt% aniline was added into the electrolyte.

## 4.4. Sample-P

Super P was used as cathode active materials, and the cathode was composed of Super P : PVDF at a weight ratio of 9 : 1 on Al foil. The electrolyte of this sample was the same as Sample-A-0.3%.

**The activation process**: the electrochemical charge/discharge tests were conducted between 3.4 and 4.2 V at room temperature, to avoid the oxidation voltage platform of PB. The electric current was set as 5 C (1 C = 170 mA g$^{-1}$) in the first 12 cycles for aniline polymerization.

## 4.5. Characterizations

The XRD pattern was measured by PANalytical B.V. with a Cu K$\alpha$ radiation test ranging from 10° to 80° in a scan rate of 8° min$^{-1}$. SEM images were obtained with a Nova NanoSEM 450, at an accelerating voltage of 25 kV. In order to fix the samples on the observation table, all the samples were pasted with conductive carbon tape as base. Raman spectrometry was performed with a LabRAM HR800, 532 light source ranging from 500 to 2500 cm$^{-1}$. FT-IR spectrometry was performed with a VERTEX 70, test ranging from 500 to 4000 cm$^{-1}$. The $N_2$ adsorption/desorption isotherms for BET surface area plot and BJH pore size distribution plot test was operated by 3H-2000PM2 micro-pore analyser at 77 K, under the inert gas condition. The EA was carried out via Elementar UNICUBE for detecting element C and element N content. The inductively coupled plasma mass spectrometry (ICP-OES) was carried out via PE Avio 200 for detecting element Na and element Fe contents. The TGA were carried out with a Diamond TG at a rate of 10°C min$^{-1}$ in $N_2$, and a heating range from 30 to 600°C. All XPS spectra were collected by using Al K$\alpha$ radiation (1486.6 eV), mono-chromatized by a couple crystal mono-chromator, yielding a focused X-ray spot at work condition 2 kV and 30 mA. The alpha hemispherical analyser was conducted in the constant condition with survey scan pass energies of 200 eV to test the integrity energy band and 50 eV in a narrow scan to electively measure the specific elements. The TEM images were obtained via a Talos F200X operated at 200 kV.

## 4.6. Electrochemical measurements

The cathode was composed of PB : Ketjen black : Super P : PVDF at a weight ratio of 7 : 1 : 1 : 1 on Al foil. The mass load of active material for each electrode is approximately 1 mg cm$^{-2}$. The electrode was dried in a vacuum oven at 70°C for 10 h. The electrolyte was composed of $NaClO_4$ dissolved in 1:1 diethyl carbonate/ethylene carbonate containing 5 wt% fluoroethylene carbonate. Whatman glass fibre was used as the separator, and Na metal was used as the counter electrode. The half coin cell was assembled as followed steps: the battery case in the bottom, then put the cathode electrode in it, then the Whatman glass fibre on the cathode electrode, then the Na metal and Ni foam, finally covered with the counter battery case. The mentioned step was operated in the Ar-filled glove box (moisture and oxygen contents were less than 1 ppm). The batteries were allowed to sit for 6 h before proceeding on for electrochemical cycling. The electrochemical charge/discharge tests were conducted between 2.0 and 4.0 V at room temperature with a Land CT2001 battery tester (Land Electronics, China). Cyclic voltammetry measurements were carried on the electrochemical workstation (CHI760E, China) with the voltage range of 2.0–4.0 V (versus Na$^+$/Na) and a scan rate of 0.1 mV s$^{-1}$. Electrochemical impedance spectroscopy tests were operated on the electrochemical workstation (CHI760E, China) over the frequency range of 0.01–100 kHz with an amplitude of 5 mV.

Data accessibility. Electronic supplementary material is provided. The datasets and supplementary file have been uploaded to Dryad: https://doi.org/10.5061/dryad.05qfttf36.

Authors' contributions. Y.L. contributed to literature search, figures, study design, data collection, data analysis and writing. L.Y. contributed to literature search, figures and data analysis. Q.L. contributed to study design and data analysis. Y.Y. contributed to revising the paper.

Competing interests. There are no conflicts to declare.

Funding. This work was funded by the National Natural Science Foundation of China (grant no. 51002054).

Acknowledgements. The authors also thank the Analytical and Testing Centre of HUST for characterization.

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
