## [Peer Review File · Royal Society Open Science]

Review History

RSOS-211092.R0 (Original submission)

Review form: Reviewer 1

Is the manuscript scientifically sound in its present form?

Yes

Are the interpretations and conclusions justified by the results?

Yes

Is the language acceptable?

Yes

Do you have any ethical concerns with this paper?

No

Have you any concerns about statistical analyses in this paper?

No

Recommendation?

Major revision is needed (please make suggestions in comments)

Comments to the Author(s)

The manuscript reported the preparation and electrochemical performance of the PBs synthesized by using AA and coating PANI. The results showed that the obtained PBs have high capacity and long cycling performance. However, some issues should need to be addressed.

1. In this work, the precursor obtained by ball milling still needs to be stirred and dissolved in deionized water to obtain precipitation. If the mixed sample without ball milling is used as the precursor as the blank comparison, it can more fully explain the improvement of the electrochemical performance of Prussian blue material by ball milling preparation method.
2. As described in this paper, the formula of Sample-A was $\text{Na}_{1.18}\text{Fe}[\text{Fe}(\text{CN})_6]_{0.795} \cdot 1.45\text{H}_2\text{O}$. Compared with the sodium rich Prussian blue synthesized by the traditional coprecipitation method, the Prussian blue material obtained by ball milling seems to have lower sodium content and higher defect content, which can be an obstacle to being used as the cathode material.
3. The small couple of peaks at 3.45/3.23 V emerges in the CV curve of Sample-A needs to be further clarified.
4. The improvement of the electrochemical properties of materials, especially the conductivity, can usually be presented in the comparison of the electrochemical impedance spectrum. The EIS of the cycling electrodes should be added to compare.

Review form: Reviewer 2

Is the manuscript scientifically sound in its present form?

No

Are the interpretations and conclusions justified by the results?

Yes

Is the language acceptable?

No

Do you have any ethical concerns with this paper?

No

Have you any concerns about statistical analyses in this paper?

No

Recommendation?

Major revision is needed (please make suggestions in comments)

Comments to the Author(s)

The authors prepared polyaniline coated-prussian blue as cathode material for sodium ion batteries (SIBs) via ball milling and rapid precipitation. The authors found improved performance when polyaniline was added to the electrolyte.

Comments to authors

- English language needs to be revised thoroughly (structurally and grammatically). Ex: (in page 5: at a high temperature is regarded). Using (considered) is more appropriate than

regarded. Also (given the instability of PB). Better to rephrase it as (due to the instability of PB). Other examples can be found by carefully revising the manuscript.

- It is not very clear what is the novelty of this work. The authors need to point out what are the advantages of the ball milling and rapid precipitation compared to the other techniques used for PB preparation.
- In the abstract the authors said that Prussian blue was used as an anode in SIBs, however, they used it as a cathode. This needs to be corrected.
- A more thorough literature review is needed of what short summary of the results summarized in table to compare it the author's findings.
- Related also to the literature review: The authors said (The Prussian blue can synthesize via a ball-milling method, which greatly improved its rate performance). Why? The authors need to mention the reason for the battery improvement when PB was prepared by ball milling. Additionally, it is important to make a comparison between the battery performance (cycling stability, capacity...etc) between this work and PB prepared by the other techniques. Authors also need to include recent papers from 2021 related to the subject.
- In the experimental section, a thorough revision is needed. In its current form the experimental part can't be duplicated by other researchers. No concentration or quantity of the materials used were reported. Some examples are as follows: The authors need to report materials' purity and manufacturer (PS: Aladdin is not the manufacturer). Also, in the sample preparation, in what ratios the chemicals were mixed? What are the ball milling time and speed?. The authors said in the sample preparation that (specific amount of ascorbic acid was used), this is not enough, what is the concentration and the volume used?. How much DI was used? Stirring time...etc all of the parameters must be reported. The preparation parameters in the current form are ambiguous and not useful for experiment duplication. The authors said that they prepared the PB by ball milling and rapid precipitation process, however, no time was reported to have an idea how fast is the process.
- It is not clear what is the difference between sample A and sample C. The authors mentioned that sample C is similar to sample A, however, anillin was added into the electrolyte. How this would make sample C different than sample A? in other words, the electrolyte is not part of the samples so adding anillin to electrolyte does not make new cathodes but makes a cell with different electrolyte. Unless the authors mean something else, then the preparation of sample C should be described in a better way.
- The authors need to report how they prepared the cells and what type? i.e. are they coin cells or Pouch cells?
- The authors prepared the cathode by coating a slurry of the active material on Al foil. This is important because aluminum does not contribute to the measured capacity as compared to steel is explained in this paper (1), and that's why aluminum is used as a substrate by researchers. It is good to address this point in the authors' work
- Abbreviations should be explained before first used (example: PANI was mentioned in the text without referring to it as polyaniline). The same also applied to PVDF and AA.
- The results' figures should be preceded by the explanation, i.e., the authors should start with the discussion then show the figures as they proceed with their discussion not the vice versa (example figure 1 was presented before the discussion).

- Why there is no SEM images for sample-O? SEM images of sample O should be provided to compare it with sample-A
- The authors reported in the results section that irregular particles led to excellent electrochemical performance. The authors may explain this by attributing it to the larger surface area of the irregular particles compared to the cubic ones. The larger surface area would in turn increase the contact area between the electrolyte and the cathode material and therefore improve battery performance. Such finding was found in this research paper (2).
- Figure 1b shows that the crystallinity (peaks intensities) of sample A is less than sample B. Why is that? though sample A showed better battery performance. It is also good to show the XRD pattern for sample A and C in one plot for comparison purposes. Additionally, It is known in the literature that the intensity of the XRD peaks increases with increasing precursor concentration, temperature, and thickness as found by others (3-9). It is good for the authors to relate this point in their work.
- Figure 2c, shows a drop in the capacity after the first few cycles which could be contributed to the formation of solid electrolyte interface (SEI) as found by other researchers (10).
- The authors prepared PB in the nanometer scale (as seen from the TEM images) which is very important for battery performance. The authors are recommended to address the importance of the size of the particles in the nanoscale size in improving battery performance as can be found in the work of other researchers (11). This is in addition to the other findings of the work in improving the battery performance.

1. Khateeb SA, Lind AG, Santos-Ortiz R, Shepherd ND, Jones KS. Effects of Steel Cell Components on Overall Capacity of Pulsed Laser Deposited FeF₂ Thin Film Lithium Ion Batteries. *Journal of The Electrochemical Society*. 2015;162(8):A1667-A74.
2. Al Khateeb S, Sparks TD. Pore-graded and conductor- and binder-free FeS₂ films deposited by spray pyrolysis for high-performance lithium-ion batteries. *Journal of Materials Research*. 2019;34(14):2456-71.
3. Shadi Al Khateeb, Button TW, Abell JS. Spray pyrolysis of MgO templates on Hastelloy C276 and 310-austenitic stainless steel substrates for Y Ba 2 Cu 3 O 7 (YBCO) deposition by pulsed laser deposition. *Superconductor Science and Technology*. 2010;23(9):095001.
4. Trofimov VI, Trofimov IV, Kim J-I. The effect of finite film thickness on the crystallization kinetics of amorphous film and microstructure of crystallized film. *Thin Solid Films*. 2006;495(1):398-403.
5. Ritala M, Leskelä M, Niinistö L, Prohaska T, Friedbacher G, Grasserbauer M. Development of crystallinity and morphology in hafnium dioxide thin films grown by atomic layer epitaxy. *Thin Solid Films*. 1994;250(1):72-80.
6. Shadi Al-Khateeb, Pavlopoulos D, Button TW, Abell JS. Pulsed Laser Deposition of YBa₂Cu₃O₇ Superconducting Film on MgO Templates Spray Pyrolyzed on Hastelloy C276. *Journal of Superconductivity and Novel Magnetism*. 2012;25(6):1823-7.
7. Pavlopoulos D, Shadi Al-Khateeb, Button TW, Abell JS. Effort to produce textured CeO₂ and MgO films by the spray pyrolysis technique as buffer layers for coated conductors. *Journal of Physics: Conference Series*. 2008;97:012098.
8. Shadi Al-Khateeb, Pavlopoulos D, Button TW, Abell JS. Spray Pyrolysis of MgO Templates on 321-Austenitic Stainless Steel Substrates for YBa₂Cu₃O₇ Deposition by PLD. *Journal of Superconductivity and Novel Magnetism*. 2013;26(2):273-80.
9. Shadi Al Khateeb, T. W. Button, J. S. Abell. Spray pyrolysis of MgO templates on Hastelloy C276 and 310-austenitic stainless steel substrates for Y Ba 2 Cu 3 O 7 (YBCO) deposition by pulsed laser deposition. *Superconductor Science and Technology*. 2010;23(9):095001.

10. Santos-Ortiz R, Rojhirunsakool T, Jha JK, Al Khateeb S, Banerjee R, Jones KS, et al. Analysis of the structural evolution of the SEI layer in FeF₂ thin-film lithium-ion batteries upon cycling using HRTEM and EELS. *Solid State Ionics*. 2017;303:103-12.
11. Shadi Al-khateeb, Lind AG, Santos-Ortiz R, Shepherd ND, Jones KS. Cycling performance and morphological evolution of pulsed laser-deposited FeF₂ thin film cathodes for Li-ion batteries. *Journal of Materials Science*. 2015;50(15):5174-82.

Decision letter (RSOS-211092.R0)

Dear Dr Luo:

Title: In situ polyaniline coating of Prussian blue as cathode material for sodium-ion battery
Manuscript ID: RSOS-211092

The editor assigned to your manuscript has now received comments from reviewers. We would like you to revise your paper in accordance with the referee and Subject Editor suggestions which can be found below (not including confidential reports to the Editor). Please note this decision does not guarantee eventual acceptance.

Please submit your revised paper before 07-Oct-2021. Please note that the revision deadline will expire at 00.00am on this date. If we do not hear from you within this time then it will be assumed that the paper has been withdrawn. In exceptional circumstances, extensions may be possible if agreed with the Editorial Office in advance. We do not allow multiple rounds of revision so we urge you to make every effort to fully address all of the comments at this stage. If deemed necessary by the Editors, your manuscript will be sent back to one or more of the original reviewers for assessment. If the original reviewers are not available we may invite new reviewers.

Yours sincerely,
Dr Ellis Wilde
Publishing Editor, Journals

On behalf of the Subject Editor Professor Anthony Stace and the Associate Editor Dr Dattatray Late.

RSC Associate Editor
Comments to the Author:
Major revision is needed

RSC Subject Editor
Comments to the Author:
(There are no comments.)

Reviewers' Comments to Author:

Reviewer: 1

Comments to the Author(s)

The manuscript reported the preparation and electrochemical performance of the PBs synthesized by using AA and coating PANI. The results showed that the obtained PBs have high capacity and long cycling performance. However, some issues should need to be addressed.

1. In this work, the precursor obtained by ball milling still needs to be stirred and dissolved in deionized water to obtain precipitation. If the mixed sample without ball milling is used as the precursor as the blank comparison, it can more fully explain the improvement of the electrochemical performance of Prussian blue material by ball milling preparation method.
2. As described in this paper, the formula of Sample-A was $\text{Na}_{1.18}\text{Fe}[\text{Fe}(\text{CN})_6]_{0.795} \cdot 1.45\text{H}_2\text{O}$. Compared with the sodium rich Prussian blue synthesized by the traditional coprecipitation method, the Prussian blue material obtained by ball milling seems to have lower sodium content and higher defect content, which can be an obstacle to being used as the cathode material.
3. The small couple of peaks at 3.45/3.23 V emerges in the CV curve of Sample-A needs to be further clarified.
4. The improvement of the electrochemical properties of materials, especially the conductivity, can usually be presented in the comparison of the electrochemical impedance spectrum. The EIS of the cycling electrodes should be added to compare.

Reviewer: 2

Comments to the Author(s)

The authors prepared polyaniline coated-prussian blue as cathode material for sodium ion batteries (SIBs) via ball milling and rapid precipitation. The authors found improved performance when polyaniline was added to the electrolyte.

Comments to authors

- English language needs to be revised thoroughly (structurally and grammatically). Ex: (in page 5: at a high temperature is regarded). Using (considered) is more appropriate than regarded. Also (given the instability of PB). Better to rephrase it as (due to the instability of PB). Other examples can be found by carefully revising the manuscript.

- It is not very clear what is the novelty of this work. The authors need to point out what are the advantages of the ball milling and rapid precipitation compared to the other techniques used for PB preparation.

- In the abstract the authors said that Prussian blue was used as an anode in SIBs, however, they used it as a cathode. This needs to be corrected.

- A more thorough literature review is needed of what short summary of the results summarized in table to compare it the author's findings.

- Related also to the literature review: The authors said (The Prussian blue can synthesize via a ball-milling method, which greatly improved its rate performance). Why? The authors need to mention the reason for the battery improvement when PB was prepared by ball milling. Additionally, it is important to make a comparison between the battery performance (cycling stability, capacity...etc) between this work and PB prepared by the other techniques. Authors also need to include recent papers from 2021 related to the subject.

- In the experimental section, a thorough revision is needed. In its current form the experimental part can't be duplicated by other researchers. No concentration or quantity of the materials used were reported. Some examples are as follows: The authors need to report materials' purity and manufacturer (PS: Aladdin is not the manufacturer). Also, in the sample preparation, in what ratios the chemicals were mixed? What are the ball milling time and speed?. The authors said in the sample preparation that (specific amount of ascorbic acid was used), this is not enough, what is the concentration and the volume used?. How much DI was used? Stirring time...etc all of the parameters must be reported. The preparation parameters in the current form are ambiguous and not useful for experiment duplication. The authors said that they prepared the PB by ball milling and rapid precipitation process, however, no time was reported to have an idea how fast is the process.

- It is not clear what is the difference between sample A and sample C. The authors mentioned that sample C is similar to sample A, however, anillin was added into the electrolyte. How this would make sample C different than sample A? in other words, the electrolyte is not part of the samples so adding anillin to electrolyte does not make new cathodes but makes a cell with different electrolyte. Unless the authors mean something else, then the preparation of sample C should be described in a better way.

- The authors need to report how they prepared the cells and what type? i.e. are they coin cells or Pouch cells?

- The authors prepared the cathode by coating a slurry of the active material on Al foil. This is important because aluminum does not contribute to the measured capacity as compared to steel

is explained in this paper (1), and that's why aluminum is used as a substrate by researchers. It is good to address this point in the authors' work

- Abbreviations should be explained before first used (example: PANI was mentioned in the text without rereferring to it as polyaniline). The same also applied to PVDF and AA.
 - The results' figures should be preceded by the explanation, i.e., the authors should start with the discussion then show the figures as they proceed with their discussion not the vise versa (example figure 1 was presented before the discussion).
 - Why there is no SEM images for sample-O? SEM images of sample O should be provided to compare it with sample-A
 - The authors reported in the results section that irregular particles led to excellent electrochemical performance. The authors may explain this by attributing it to the larger surface area of the irregular particles compared to the cubic ones. The larger surface area would in turn increase the contact area between the electrolyte and the cathode material and therefore improve battery performance. Such finding was found in this research paper (2).
 - Figure 1b shows that the crystallinity (peaks intensities) of sample A is less than sample B. Why is that? though sample A showed better battery performance. It is also good to show the XRD pattern for sample A and C in one plot for comparison purposes. Additionally, It is known in the literature that the intensity of the XRD peaks increases with increasing precursor concentration, temperature, and thickness as found by others (3-9). It is good for the authors to relate this point in their work.
 - Figure 2c, shows a drop in the capacity after the first few cycles which could be contributed to the formation of solid electrolyte interface (SEI) as found by other researchers (10).
- The authors prepared PB in the nanometer scale (as seen from the TEM images) which is very important for battery performance. The authors are recommended to address the importance of the size of the particles in the nanoscale size in improving battery performance as can be found in the work of other researchers (11). This is in addition to the other findings of the work in improving the battery performance.
1. Khateeb SA, Lind AG, Santos-Ortiz R, Shepherd ND, Jones KS. Effects of Steel Cell Components on Overall Capacity of Pulsed Laser Deposited FeF₂ Thin Film Lithium Ion Batteries. *Journal of The Electrochemical Society*. 2015;162(8):A1667-A74.
 2. Al Khateeb S, Sparks TD. Pore-graded and conductor- and binder-free FeS₂ films deposited by spray pyrolysis for high-performance lithium-ion batteries. *Journal of Materials Research*. 2019;34(14):2456-71.
 3. Shadi Al Khateeb, Button TW, Abell JS. Spray pyrolysis of MgO templates on Hastelloy C276 and 310-austenitic stainless steel substrates for Y Ba₂ Cu₃ O₇ (YBCO) deposition by pulsed laser deposition. *Superconductor Science and Technology*. 2010;23(9):095001.
 4. Trofimov VI, Trofimov IV, Kim J-I. The effect of finite film thickness on the crystallization kinetics of amorphous film and microstructure of crystallized film. *Thin Solid Films*. 2006;495(1):398-403.
 5. Ritala M, Leskelä M, Niinistö L, Prohaska T, Friedbacher G, Grasserbauer M. Development of crystallinity and morphology in hafnium dioxide thin films grown by atomic layer epitaxy. *Thin Solid Films*. 1994;250(1):72-80.
 6. Shadi Al-Khateeb, Pavlopoulos D, Button TW, Abell JS. Pulsed Laser Deposition of YBa₂Cu₃O₇ Superconducting Film on MgO Templates Spray Pyrolyzed on Hastelloy C276. *Journal of Superconductivity and Novel Magnetism*. 2012;25(6):1823-7.

7. Pavlopoulos D, Shadi Al-Khateeb, Button TW, Abell JS. Effort to produce textured CeO₂ and MgO films by the spray pyrolysis technique as buffer layers for coated conductors. *Journal of Physics: Conference Series*. 2008;97:012098.
8. Shadi Al-Khateeb, Pavlopoulos D, Button TW, Abell JS. Spray Pyrolysis of MgO Templates on 321-Austenitic Stainless Steel Substrates for YBa₂Cu₃O₇ Deposition by PLD. *Journal of Superconductivity and Novel Magnetism*. 2013;26(2):273-80.
9. Shadi Al Khateeb, T. W. Button, J. S. Abell. Spray pyrolysis of MgO templates on Hastelloy C276 and 310-austenitic stainless steel substrates for Y Ba 2 Cu 3 O 7 (YBCO) deposition by pulsed laser deposition. *Superconductor Science and Technology*. 2010;23(9):095001.
10. Santos-Ortiz R, Rohirunsakool T, Jha JK, Al Khateeb S, Banerjee R, Jones KS, et al. Analysis of the structural evolution of the SEI layer in FeF₂ thin-film lithium-ion batteries upon cycling using HRTEM and EELS. *Solid State Ionics*. 2017;303:103-12.
11. Shadi Al-khateeb, Lind AG, Santos-Ortiz R, Shepherd ND, Jones KS. Cycling performance and morphological evolution of pulsed laser-deposited FeF₂ thin film cathodes for Li-ion batteries. *Journal of Materials Science*. 2015;50(15):5174-82.

Author's Response to Decision Letter for (RSOS-211092.R0)

See Appendix A.

Decision letter (RSOS-211092.R1)

Dear Dr Luo:

Title: In situ polyaniline coating of Prussian blue as cathode material for sodium-ion battery
Manuscript ID: RSOS-211092.R1

It is a pleasure to accept your manuscript in its current form for publication in Royal Society Open Science. The chemistry content of Royal Society Open Science is published in collaboration with the Royal Society of Chemistry.

Yours sincerely,
Dr Ellis Wilde

Publishing Editor, Journals

On behalf of the Subject Editor Professor Anthony Stace and the Associate Editor Dr Dattatray Late.

RSC Associate Editor
Comments to the Author:
Accept as is

Reviewer(s)' Comments to Author:

Appendix A

Dear Editor and Reviewers:

Thank you for your letter and for the reviewers' comments concerning our manuscript entitled "In situ polyaniline coating of Prussian blue as cathode material for sodium-ion battery" (Manuscript ID: RSOS-211092). Those comments are all valuable and very helpful for revising and improving our paper. We have studied the comments carefully and have made corrections which we hope meet with the requirements. Revised portions are marked in red in the paper. The main corrections in the paper and the responses to the reviewer's comments are listed in the following:

Responses to the reviewer's comments:

Reviewer 1:

1. In this work, the precursor obtained by ball milling still needs to be stirred and dissolved in deionized water to obtain precipitation. If the mixed sample without ball milling is used as the precursor as the blank comparison, it can more fully explain the improvement of the electrochemical performance of Prussian blue material by ball milling preparation method.

Response: We appreciate for your comments. We set the Sample-L for the comparison sample without ball milling is used as the precursor, directly by a rapid co-precipitation to synthesis Prussian blue. And the characterizations and electrochemical test (XRD, FT-IR, TG, BET, BJH, electrochemical cycling performance, *etc.*) related to the Sample-L have been added in the figures. We made the FT-IR measurement to get rid of Raman test because that it is more sensitive for the detection of organic functional groups by the FT-IR test compared with Raman test.

2. As described in this paper, the formula of Sample-A was $\text{Na}_{1.18}\text{Fe}[\text{Fe}(\text{CN})_6]_{0.795} \cdot 1.45\text{H}_2\text{O}$. Compared with the sodium rich Prussian blue synthesized by the traditional coprecipitation method, the Prussian blue material obtained by ball milling seems to have lower sodium content and higher defect content, which can be an obstacle to being used as the cathode material.

Response: We deeply appreciate your comments. The formula of Sample-A presented in the original manuscript is imperfectly accurate. This phenomenon may be caused by the un-complete dissolution of the Prussian blue in the ICP test preparatory stage in our lab. In the past month, we sent the samples to a more specialized laboratory in our institute and collected the correct information. We have renewed the ICP/EA test result (the machine types relate to ICP/EA test are also change) and calculate the formula in the new submitted article. The formula of Sample-A is denoted as $\text{Na}_{1.515}\text{Fe}[\text{Fe}(\text{CN})_6]_{0.879} \cdot 2.7\text{H}_2\text{O}$. Then we list the formulas of Prussian blues recently reported in other article, as shown in the Fig. S12 (in the figure we could conclude the sodium content and defect content of the Prussian blue that other literature reported).

3. The small couple of peaks at 3.45/3.23 V emerges in the CV curve of the Sample-A needs to be further clarified.

Response: Thanks for your comments. The small couple of peaks at 3.45/3.23 V emerges in the CV curve of the Sample-A might be caused by the extraction/insertion of Na^+ from some other sites in

the structure, related to the N-Fe^{III}. We have clarified this point with the discussion of XPS spectrum (Figure S1). Besides, other literatures also give the evidence, as listed in the followed paragraph:

i. Wanlin Wang, Yong Gang, Zhe Hu, Zichao Yan, Weijie Li, Yongcheng Li, Qin-Fen Gu, Zhixing Wang, Shu-Lei Chou, Hua-Kun Liu, and Shi-Xue Dou. Reversible structural evolution of sodium-rich rhombohedral Prussian blue for sodium-ion batteries. *NATURE COMMUNICATIONS*. 2020, 11, 980.

In the page 4

The peaks at 2.8–3.1 V are related to the redox reactions of high spin Fe²⁺ connected with N atoms, and peaks round ~3.8 V corresponds to the low spin Fe²⁺ connected with C atoms, while the oxidization peak for PB-S1 over 3.8 V is more obvious than that of PB-S3, which is consistent with the higher long plateau in the charge curve of PB-S1 (Fig. 3a), the reason might due to the deficient Fe²⁺ from N sites in the cubic phase, more Na⁺ would be extracted during charge process by oxidization of Fe²⁺ from C sites. While the small redox peaks around 3.3–3.4 V for both sample might be caused by Na⁺ extracted/inserted from some other sites in the structure.

ii. L. Li, P. Nie, Y. B. Chen and J. Wang. Novel acetic acid induced Na-rich Prussian blue nano cubes with iron defects as cathodes for sodium ion batteries. *J. Mater. Chem. A*. 2019, 7, 12134–12144.

In the page 12139 paragraph 6

According to previous work, two pairs of redox peaks at 3.13/2.73 V and 3.34/3.25 V for the PB-1 electrode are attributed to the high-spin Fe²⁺/Fe³⁺ couple coordinated to the N atoms of (C-N).

4. The improvement of the electrochemical properties of materials, especially the conductivity, can usually be presented in the comparison of the electrochemical impedance spectrum. The EIS of the cycling electrodes should be added to compare.

Response: We appreciate for your comments. The EIS results for electrodes with different electrochemical cycles are shown in the Figure S6. The gradually changed radius of the semicircle (relate to the Rct) represents the gradually enhanced conductivity. The charge transfer kinetics of the electrodes benefitted from the conductive polymer coating and a more stable interface between the electrode & electrolyte in the running cycles. Similar phenomenon have also been observed in the literature on the effect of the additives in the electrolyte reported before as listed in the following.

i. Fangyuan Cheng, Xiaoyu Zhang, Yuegang Qiu, Jinxu Zhang, Yi Liu, Peng Wei, Mingyang Ou, Shixiong Sun, Yue Xu, Qing Li, Chun Fang, Jiantao Han and Yunhui Huang. Tailoring electrolyte to enable high-rate and super-stable Ni-rich NCM cathode materials for Li-ion batteries. *Nano Energy*. 2021, 88, 106301.

In the page 7

R_s represents the solution impedance, R_f = (R_{i1} + R_{i2}) represents SEI/CEI impedance and R_{ct} represents the charge transfer impedance across the particle surface. After the 10th cycle, the resistance of the cells with or without additives decreased significantly due to the formation of CEI and SEI. As the cycle number increasing, the impedance of the cells with baseline electrolyte

increased gradually, as shown in Fig. 5a, owing to the formation of a continuously growing thick interphase. However, in the additives-containing electrolyte, the impedance of the cells is more stable and smaller than those in the baseline electrolyte, especially for the sample with dual additives of LiBOB + DA (Fig. 5b–d). The comparison of impedance trends is shown in Fig. S18 (Supporting information). Combined with Fig. 5b, it is clear that the N-rich dopamine derived CEI has a much lower impedance, which promotes the diffusion of Li⁺ due to the high affinity of N to Li⁺ ions.

ii. **Yang Liu, Dandan He, Yingjie Cheng, Lin Li, Zhansheng Lu, Rui Liang, Yangyang Fan, Yun Qiao, and Shulei Chou. A Heterostructure Coupling of Bioinspired, Adhesive Polydopamine, and Porous Prussian Blue Nanocubics as Cathode for High-Performance Sodium-Ion Battery. *Small*. 2020, 16, 1906946.**

In the page 4

Electrochemical impedance spectra (EIS) further indicate that the NFF@PDA electrode presents relatively low contact, charge-transfer resistance and diffusion impedances, in contrast to NFF cathode (Figure S8, Supporting Information).

Special thanks to you for your good comments.

Reviewer 2:

1. English language needs to be revised thoroughly (structurally and grammatically). Ex: (in page 5: at a high temperature is regarded). Using (considered) is more appropriate than regarded. Also (given the instability of PB). Better to rephrase it as (due to the instability of PB). Other examples can be found by carefully revising the manuscript.

Response: We appreciate for your advices. We have corrected the mistakes and improved the English writing in the newly provided manuscript.

2. It is not very clear what is the novelty of this work. The authors need to point out what are the advantages of the ball milling and rapid precipitation compared to the other techniques used for PB preparation.

Response: We appreciate for your advices. We point out the advantages of the ball milling and rapid precipitation in the new provided article. In the end of introduction part: 1. compared to the traditional synthesis method, the Prussian blue synthesized in our method had lower interstitial water content. 2. The rate electrochemical performance of the Prussian blue produced by our two steps synthesis method greatly improved compared to the traditional synthesis method, owing to the large surface area (correspond to the BET plot in figure S4) 3. Ascorbic acid (AA) can prevent the oxidation of Prussian blue (Fe²⁺ changes into Fe³⁺) during the ball mill process to some extent, which lead to the richer sodium ion source (corresponding to the XPS spectrum in figure S1). We compared the advantages of our method to the other reported techniques in the Figure S12 (The rate electrochemical performance and the times cost in the synthesis process were list in Figure S12).

3. In the abstract the authors said that Prussian blue was used as an anode in SIBs, however, they used it as a cathode. This needs to be corrected.

Response: Thanks for your comments. In the submitted revised manuscript, we have corrected the mistake.

4. A more thorough literature review is needed of what short summary of the results summarized in table to compare it the author's findings.

Response: We deeply appreciate your comments. A more thorough literature review (in recently two years) is provided in our submitted revised manuscript, as show in the figure S12.

5. Related also to the literature review: The authors said (The Prussian blue can synthesize via a ball-milling method, which greatly improved its rate performance). Why? The authors need to mention the reason for the battery improvement when PB was prepared by ball milling. Additionally, it is important to make a comparison between the battery performance (cycling stability, capacity...etc) between this work and PB prepared by the other techniques. Authors also need to include recent papers from 2021 related to the subject.

Response: We appreciate for your advices. The Prussian blue can be synthesized via a ball-milling method, which greatly improves its rate performance owing to the large surface area (correspond to the BET plot in figure S4). And we compared the advantages of our method to the other techniques (recent papers from 2021 related to the subject) in the Figure S12 (The electrochemical performance were list in Figure S12).

6. In the experimental section, a thorough revision is needed. In its current form the experimental part can't be duplicated by other researchers. No concentration or quantity of the materials used were reported. Some examples are as follows: The authors need to report materials' purity and manufacturer (PS: Aladdin is not the manufacturer). Also, in the sample preparation, in what ratios the chemicals were mixed? What are the ball milling time and speed?. The authors said in the sampoe preparation that (specific amount of ascorbic acid was used), this is not enough, what is the concentration and the volume used?. How much DI was used? Stirring time...etc all of the parameters must be reported. The preparation parameters in the current form are ambiguous and not useful for experiment duplication. The authors said that they prepared the PB by ball milling and rapid precipitation process, however, no time was reported to have an idea how fast is the process.

Response: We appreciate for your advices. We have improved this part (The Experimental Section) in the new submit manuscript.

7. It is not clear what is the difference between sample A and sample C. The authors mentioned that sample C is similar to sample A, however, anillin was added into the electrolyte. How this would make sample C different than sample A? in other words, the electrolyte is not part of the

samples so adding anillin to electrolyte does not make new cathodes but makes a cell with different electrolyte. Unless the authors mean something else, then the preparation of sample C should be described in a better way.

Response: We appreciate for your advices and feel sorry for making the reviewer confused. The Sample C's cathode electrode is the same as that used for Sample-A, but 0.3 wt. % aniline was added into the electrolyte for conductive polymer coating. We used the new name Sample-A-0.3% to replace the old name Sample C, and the Characterizations and Electrochemical test related to Sample C made the same name change. Additionally, we change the Sample-B name to Sample-A-3% in the same way. We have corrected this part in the new submit manuscript.

8. The authors need to report how they prepared the cells and what type? i.e. are they coin cells or Pouch cells?⁴

Response: We appreciate for your advices. We have improved this part (The Experimental Section) in the newly submitted manuscript. They are coin cells.

9. The authors prepared the cathode by coating a slurry of the active material on Al foil. This is important because aluminum does not contribute to the measured capacity as compared to steel is explained in this paper (1), and that's why aluminum is used as a substrate by researchers. It is good to address this point in the authors' work

Response: We deeply appreciate your comments. We agree with this point and have cited this paper in the newly submitted manuscript. The reference number in our new submit article is 29.

10. Abbreviations should be explained before first used (example: PANI was mentioned in the text without rereferring to it as polyaniline). The same also applied to PVDF and AA.

Response: Thanks for reviewer's advice. We have improved this point in the newly submitted manuscript. The abbreviations now are explained before first used.

11. The results' figures should be preceded by the explanation, i.e., the authors should start with the discussion then show the figures as they proceed with their discussion not the vise versa (example figure 1 was presented before the discussion).

Response: We appreciate for your advices. We start with the discussion then show the correspond figures. We have improved this point in the new submit manuscript.

12. Why there is no SEM images for sample-O? SEM images of sample O should be provided to compare it with sample-A

Response: We appreciate for your advices. We provide the SEM images of all the samples (figure S2) and the related TEM images (figure S3), in the newly submitted manuscript.

13. The authors reported in the results section that irregular particles led to excellent electrochemical performance. The authors may explain this by attributing it to the larger surface area of the irregular particles compared to the cubic ones. The larger surface area would in turn increase the contact area between the electrolyte and the cathode material and therefore improve battery performance. Such finding was found in this research paper (2).

Response: We deeply appreciate your comments. We agree with this point and have cited the paper in the new submit manuscript. The reference number in our new submit article is 20. Additionally, the Brunner Emmet Teller (BET) plot and the Barrett Joyner Halenda (BJH) plot of all the samples are provided (figure S4 and figure S5).

14. Figure 1b shows that the crystallinity (peaks intensities) of sample A is less than sample B. Why is that? though sample A showed better battery performance. It is also good to show the XRD pattern for sample A and C in one plot for comparison purposes. Additionally, It is known in the literature that the intensity of the XRD peaks increases with increasing precursor concentration, temperature, and thickness as found by others (3-9). It is good for the authors to relate this point in their work.

Response: Thanks for reviewer advices. We agree with the point that the intensity of the XRD peaks increases with increasing precursor concentration, temperature, and thickness as found by others. The crystallinity (peaks intensities) of sample-A is less than sample-O. This phenomenon may owe to the different mass weight of the samples add into the X-ray diffraction test. As Reviewer 1 suggested and made a new sample (the Sample-L). In the new submit article we made the fresh XRD test and get the correct information (figure 1). In the Question 7 we explain that the Sample C's cathode electrode is the same as that used for the Sample-A, and change the Sample C name to the Sample-A-0.3%. The XRD plot of the old Sample C is the same with the Sample-A. Additionally, we agree with the reviewer points and add them in the new submit manuscript. The reference numbers in our new submit article are 12-18.

15. Figure 2c, shows a drop in the capacity after the first few cycles which could be contributed to the formation of solid electrolyte interface (SEI) as found by other researchers (10).

Response: We deeply appreciate your comments. We agree with this point and add it in the new submit manuscript. The reference number in our new submit article is 22.

16. The authors prepared PB in the nanometer scale (as seen from the TEM images) which is very important for battery performance. The authors are recommended to address the importance of the size of the particles in the nanoscale size in improving battery performance as can be found in the work of other researchers (11). This is in addition to the other findings of the work in improving the battery performance.

Response: We greatly appreciate for your comments. We agree with this point and add it in the new submit manuscript. The reference number in our new submit article is 19.

Special thanks to you for your good comments.

We tried our best to improve the manuscript and made some changes in the manuscript. These changes will not influence the content and framework of the paper. And here we did not list the changes but marked in red in revised paper.

We appreciate for Editors/Reviewers' warm work earnestly, and hope that the correction will meet with approval.

Once again, thank you very much for your comments and suggestions.

Yours Sincerely,
Youwei Yan.